# Short Arrestin-3-Derived Peptides Activate JNK3 in Cells

**DOI:** 10.3390/ijms23158679

**Published:** 2022-08-04

**Authors:** Nicole A. Perry-Hauser, Tamer S. Kaoud, Henriette Stoy, Xuanzhi Zhan, Qiuyan Chen, Kevin N. Dalby, Tina M. Iverson, Vsevolod V. Gurevich, Eugenia V. Gurevich

**Affiliations:** 1Department of Pharmacology, Vanderbilt University, Nashville, TN 37232, USA; 2Division of Chemical Biology & Medicinal Chemistry, The University of Texas at Austin, Austin, TX 78712, USA; 3Institute of Molecular Cancer Research, University of Zurich, Ramistrasse 71, CH-8006 Zurich, Switzerland; 4Center for Structural Biology, Vanderbilt University, Nashville, TN 37232, USA; 5Vanderbilt Institute of Chemical Biology, Vanderbilt University, Nashville, TN 37232, USA; 6Department of Biochemistry, Vanderbilt University, Nashville, TN 37232, USA

**Keywords:** arrestin-3, scaffold, JNK, short peptides

## Abstract

Arrestins were first discovered as suppressors of G protein-mediated signaling by G protein-coupled receptors. It was later demonstrated that arrestins also initiate several signaling branches, including mitogen-activated protein kinase cascades. Arrestin-3-dependent activation of the JNK family can be recapitulated with peptide fragments, which are monofunctional elements distilled from this multi-functional arrestin protein. Here, we use maltose-binding protein fusions of arrestin-3-derived peptides to identify arrestin elements that bind kinases of the ASK1-MKK4/7-JNK3 cascade and the shortest peptide facilitating JNK signaling. We identified a 16-residue arrestin-3-derived peptide expressed as a Venus fusion that leads to activation of JNK3α2 in cells. The strength of the binding to the kinases does not correlate with peptide activity. The ASK1-MKK4/7-JNK3 cascade has been implicated in neuronal apoptosis. While inhibitors of MAP kinases exist, short peptides are the first small molecule tools that can activate MAP kinases.

## 1. Introduction

Arrestins were discovered for their role in the desensitization and internalization of active, phosphorylated G protein-coupled receptors (GPCRs) [1]. However, this is only one mechanism through which arrestins modulate signaling. The two non-visual arrestins (arrestin-2 and arrestin-3, also known as β-arrestin1 and 2, respectively) collectively interact with hundreds of signaling and trafficking proteins [2,3,4,5,6], some of which are specific for an individual subtype [7]. Importantly, arrestins facilitate the activation of kinases that determine cell fate via pro-survival or pro-apoptotic signaling. The activity of these kinases is important for organ and tissue development and has been implicated in cancer [8,9]. Understanding the mechanisms of arrestin-dependent activation of these signaling pathways will pave the way for the development of molecular tools that can modulate signaling in a specific pathway for the purpose of research and, ultimately, as therapeutics.

The scaffolding function of arrestins is well-characterized. Non-visual arrestins recruit three-tiered mitogen-activated protein kinase (MAPK) cascades that activate extracellular signal-regulated kinase (ERK1/2) [10], p38 [11], and c-Jun N-terminal kinase (JNK) [12]. MAPK signaling cascades are organized by scaffolding proteins, which recruit the kinases of all three levels (MAP3K, MAP2K, MAPK), thereby bringing them into proximity and facilitating activation by sequential phosphorylation [13,14,15]. Therefore, a plausible model of arrestin-facilitated MAPK signaling includes the recruitment of all the kinases of a particular cascade to arrestin as a necessary, but not sufficient, step [10,12].

Here, we focus on the ASK1-MKK4/7-JNK3 cascade, which is implicated in neuronal development and neurodegeneration [13,16]. It is activated by only one isoform of arrestin, arrestin-3 [12,17]. It was originally reported that arrestin-3 activated the JNK3 isoform, which is enriched in the nervous system [18]. We showed that arrestin-3 also activated ubiquitous JNK1/2 [19], although JNK3 appears to be its preferred target. We previously showed that JNK3α2 directly binds both receptor-bound and free arrestin-3 [20] and identified the first 25 N-terminal residues (T1A peptide) as critical for JNK3α2 binding and its activation in cells [21].

T1A is an element that does not have other known functions of a remarkably multi-functional arrestin-3 protein [22]. The use of monofunctional elements enables studies of individual arrestin functions [23]. Here, we investigate smaller elements of T1A and show that a 16-residue peptide termed T16 is sufficient for strong activation of the cascade. A 14-residue peptide (T14) also consistently induced robust activation of JNK3, albeit less effectively than T16. These peptides are the smallest known molecular scaffolds for the ASK1-MKK4/7-JNK3 cascade.

## 2. Results

To further probe the functional role of T1A residues in the binding of kinases of the ASK1-MKK4/7-JNK3 cascade and JNK3 activation, we constructed several shorter peptides derived from T1A. We used peptides of variable length, which spanned from Thr7 to Gly24 of the parental arrestin-3 (Figure 1). We used an arrestin-2-derived, 24-residue peptide (termed B1A), which is homologous to T1A and binds JNK3α2, but does not facilitate JNK3α2 activation in cells [21]; therefore, this peptide served as a biologically relevant negative control in all experiments. We expressed shorter T1A-derived peptides in *E. coli* as maltose binding protein (MBP) fusions and purified these constructs on an amylose column, as described previously [20].

We used purified fusion proteins to compare the binding of T1A and shorter T1A-derived peptides to each kinase in the ASK1-MKK4/7-JNK3 cascade. The binding of ASK1 to T1A was previously tested only in cells, as purified ASK1 was not available [21]. For the pull-down experiments, we were able to purify ASK1 and demonstrate that it binds with high affinity to T1A. However, pull-down did not detect its binding to full-length arrestin-3 and other peptides (Figure 2). The absence of detectable binding to full-length arrestin-3 suggests that either residues outside of the T1A element inhibit ASK1 binding or the conformation of T1A peptide within the full-length arrestin-3 is suboptimal for ASK1 binding, whereas T1A fused to MBP does not have the same conformational constraints. Conceivably, the unconstrained T1A peptide assumes the conformation conducive to ASK1 binding upon encountering ASK1 (via a well-established mechanism of interaction-induced folding [24,25]). Another aspect illustrates the limitations of pull-down: only relatively strong interactions survive washing steps, and apparently, full-length arrestin-3 binds ASK1 only transiently, even though there is strong experimental support for the ASK1-arrestin-3 interaction and its biological significance [12,17]. The binding of T16 to ASK1 is also undetectable by pull-down, even though it needs to bind ASK1 to facilitate JNK3 activation in cells (see below).

MKKs appear to bind arrestin-3 less avidly in cells than ASK1 and JNK3 [12]. It has been shown previously that MKK7 has a lower affinity for arrestin-3 than MKK4, whereas T1A has a lower affinity for MKK4 than MKK7 [27]. Arrestin-3 demonstrates higher binding to MKK4, while T1A binds MKK7 more avidly (Figure 3 and Figure 4). While the binding of T14 to MKK4 is detectable, the binding of the T1A and other shorter peptides could not be reliably detected (Figure 3). Among the short T1A-derived peptides T14 exhibits the greatest binding to MKK4. This suggests that the Val–Phe sequence present in T16 and T15 but absent in T14 (Figure 1) might act as a “brake”, facilitating MKK4 dissociation. Considering that MKK4 and MKK7, both of which are needed for full activation of JNK3, appear to compete for arrestin-3 [27], this “brake” might promote the exchange of the two MKKs on arrestin-3 [28], thereby facilitating arrestin-3-dependent JNK3 activation. Only arrestin-3 and T1A peptide demonstrated detectable pull-down binding to MKK7 (Figure 4). Full-length arrestin-3 demonstrates a robust binding to both MKKs, stronger than the best-binding short peptides (Figure 3 and Figure 4). This suggests that the MKK-binding element(s) in full-length arrestin-3 are accessible, and exist in conformation close to optimal for these interactions.

JNK3α2 is the final effector in the cascade. It demonstrated robust binding to both full-length arrestin-3 and T16, while its binding to T1A and other shorter peptides was too weak to be detected by pull-down (Figure 5). Avid binding of full-length arrestin-3 to JNK3 is consistent with in-cell data [17,29]. T14 contains the last 14 residues of T16, missing only the Val–Phe pair at the beginning (Figure 1). Higher binding to T16 than to T14 suggests that the two residues that appeared to reduce the binding of MKK4 and MKK7 (Figure 3 and Figure 4) enhance JNK3 binding (Figure 5).

Numerous protein–protein interactions critical for cell signaling have relatively low affinity, and therefore are transient in cells. These cannot be detected by in vitro pull-down. Therefore, we tested whether the T1A-derived peptides promote JNK3α2 activation in cells (Figure 6). To this end, we co-expressed HA-JNK3 and Venus-fusions of the peptides in HEK293 cells (WT) and arrestin-2/3 knockout HEK293 (DKO) cells [30,31], using Venus as a negative control and Venus-arrestin-3 as a positive control (Figure 6). The pattern of peptide activity was generally similar in WT and DKO cells (Figure 6), suggesting that their effects are direct and do not depend on the presence of endogenous arrestins. Of the T1A-derived peptides, T16 facilitated JNK3α2 activation to a greater extent than T1A, comparable to that achieved with full-length arrestin-3. Removal of amino acids to shorten T16 led to a loss of activity to a varying degree. However, both shorter peptides T15 and T14 in WT cells and T14 in DKO cells still displayed activity comparable to T1A. Only the shortest peptide, T13, was ineffective in both cell types. Collectively, these data suggest that a much shorter version of T1A, T16, is sufficient for scaffolding the ASK1-MKK4/7-JNK3 cascade resulting in JNK3 activation in cells. In both cell types tested, T16 was significantly more effective in JNK3α2 activation than a longer T1A described previously [21].

## 3. Discussion

We have recently demonstrated that a 25 amino acid N-terminal fragment of arrestin-3 termed T1A binds JNK3 and upstream kinases MKK4 and MKK7, facilitating JNK3 activation in cells [21]. Here, we screened shorter peptides derived from T1A to probe the interaction sites between arrestin-3 and kinases of the ASK1-MKK4/7-JNK3 cascade to identify minimal elements necessary and sufficient to facilitate the JNK3 activation in cells. We showed that peptide fragments as short as 14-residue T14 could facilitate JNK3 activation when expressed in cells. Furthermore, we demonstrate that a 16-residue T16 is a very effective JNK3 activator, more effective than T1A, and almost as effective as the parental full-length arrestin-3. Considering the length of the peptides, this finding might appear surprising. However, in a fully extended conformation of the peptide, each residue adds 3.8 Å. This makes the maximum length of 25- and 16-residue peptides ~95 Å and 60 Å, respectively. For comparison, the longest axis in the structure of full-length arrestins is 70–75 Å. It is also conceivable that the peptides, as well as full-length arrestin-3, facilitate signaling by tethering two kinases at any given time. While MAPK inhibitors are available and widely used, the T16 peptide described here is the first small molecule capable of efficiently activating JNK3 signaling. Selective activation of anti-proliferative, often pro-apoptotic JNK family kinases has therapeutic potential in disorders associated with excessive proliferation, such as cancer.

Mechanistically, the comparison of the ability of arrestin-3 and short peptides derived from it to bind the four kinases of the ASK1-MKK4/7-JNK3 signaling cascade (Figure 2, Figure 3, Figure 4 and Figure 5) suggests that the “best” peptides bind MKK4, MKK7, and JNK3, albeit the efficacy is mostly lower than that of the full-length protein. Surprisingly, the interaction of arrestin-3 with ASK1, the upstream-most kinase that initiates signaling, is weaker than that of the T1A fragment. It is tempting to speculate that this makes ASK1-initiated signaling in cells more dynamic. Transient low-affinity interaction with ASK1 would also allow an easy switch between different upstream MAPK kinase kinases (MAP3Ks), some of which arrestin-3 was reported to bind [10]. Another surprise was a virtually undetectable binding of smaller peptides to ASK1 (Figure 2), despite their ability to activate JNK3 in cells (Figure 6). This also suggests that arrestin-3 and, possibly, arrestin-3-derived peptides could interact with other MAP3Ks. Mammals express more different MAP3Ks than downstream kinases [33], and available data suggest that the specificity of the signaling is determined by the MAP3K involved rather than by the downstream kinases [33]. Only the binding of arrestin-3 to ASK1 [12] and of both non-visual arrestins to cRaf (the MAP3K that initiates pro-survival ERK1/2 activation cascade) [10,34] has been demonstrated experimentally. In a comprehensive survey of the interactome of the two non-visual arrestins that identified more than 100 partners, only two MAP3Ks were detected: both arrestin-2 and -3 in stimulated cells bound to MEKK1, and arrestin-3 also interacted with TAK1 [7]. This survey was based on co-immunoprecipitation, which is prone to miss lower affinity interactions such as the pull-down used here. That likely explains why neither cRaf, nor ASK1, both of which were shown to bind arrestins [10,12,17,35,36], were detected [7]. Thus, a targeted search for arrestin-binding MAP3Ks, possibly using more sensitive methods, appears to be in order.

As far as MKKs and JNK3 are concerned, T15 demonstrates little to no binding, whereas T14, T1A, and T16 show detectable binding to some of the kinases of ASK1-MKK4/7-JNK3 module (Figure 2, Figure 3, Figure 4 and Figure 5). Only T1A demonstrated detectable binding to two (out of four) kinases (Figure 2 and Figure 4). Yet functionally, all peptides except the shortest T13 facilitate JNK3 activation in cells (Figure 6). These results suggest that there is no clear correlation between the affinity of the binding and functional activity, i.e., in a multi-protein signaling complex, the strength of interaction of the scaffold with individual components cannot predict the magnitude of the resulting signaling. Indeed, while in terms of the strength of binding to ASK1 (Figure 2) and both MKKs (Figure 3 and Figure 4) T16 peptide does not stand out, it is the most effective facilitator of JNK3 activation among the peptides tested, comparable to the full-length arrestin-3 (Figure 6). The strong support for the notion that the binding of kinases to arrestin per se does not necessarily lead to activation of the kinase cascade comes from the fact that both non-visual arrestins bind the kinases of the ASK1-MKK4/7-JNK3 cascade, but only arrestin-3 facilitates the activation of JNK3 [12,17,29,37]. Thus, both binding and optimal relative orientation of the kinases in the complex are critical for the activation of MAPKs.

Interestingly, the Val–Phe residues present in T16 and absent in T14 (Figure 1) appear to serve as a “brake,” reducing the binding to MKK4. This might improve the functional efficiency of T16 in two ways. One, MKK4 and MKK7, the activity of both of which is necessary for the full JNK3 activation [38] appear to compete for the same site on arrestin-3 [27], so transient low-affinity binding might facilitate their exchange on the arrestin-3 scaffold. It would likely facilitate the release of the activated JNK3, freeing the place for another molecule of inactive JNK3, thereby increasing signal amplification [28].

The interactions of T16 with ASK1 and both MKKs are too weak to be detected by pull-down (Figure 2, Figure 3 and Figure 4), making them transient in the cell. They are sufficient for T16 to significantly facilitate JNK3 activation in cells (Figure 6). To the best of our knowledge, this peptide is the smallest described effective molecular scaffold of the ASK1-MKK4/7-JNK3 cascade or any MAPK cascade. This peptide does not have most elements involved in binding GPCRs (reviewed in [22,39]), ERK2 and its upstream activators [34,40], Src family kinases [41], clathrin [3], clathrin adaptor AP2 [5], calmodulin [42], and other known arrestin partners. Our finding suggests that other relatively small arrestin elements might be largely responsible for individual scaffolding functions. Such elements could be identified, and their ability to regulate cellular signaling in a targeted manner could thus be exploited.

Further experimentation with arrestin peptides exposed on the surface of a folded protein is needed to address this issue. In the case of the JNK activation, the critical function appears to be performed by a contiguous peptide. In other cases, creating monofunctional tools might require stitching together two or more peptides from different parts of the arrestin surface. This would make this research direction harder but will not make it impossible.

Signaling abnormalities caused by faulty protein–protein interactions often underlie diseases [43,44,45]. In many disease cases, multiple protein–protein interactions are perturbed by mutations and would require corrections. Signaling via protein–protein interactions is amenable to interruption resulting in inhibition, whereas the correction could rarely be achieved if the activation is required. Furthermore, correcting signaling abnormalities for therapeutic purposes is complicated by the multifunctional nature of most proteins (arrestin-3 is a case in point [22,23]), which is needed to regulate cellular signaling in a coordinated manner. Therapeutic intervention to correct disease-causing signaling errors would require selective manipulation of individual signaling branches without affecting others. Protein elements that retain a single function of the parental protein and selectively activate only one signaling pathway in the cell are the molecular tools that could be used for research and eventually exploited for therapeutic purposes. For example, peptide mini-scaffolds of the JNK cascade described here can be used as highly selective molecular tools to direct signaling toward a specific outcome.

## 4. Materials and Methods

### 4.1. Protein Purification

Arrestin was expressed in *E. coli* cells and purified, as described [46]. JNK3α2 and MKK4/7 were expressed and purified, as described in [19,20,27], and ASK1, as described in [47]. The MBP fusions of arrestin-3 and peptides were purified on an amylose column, as described [20].

### 4.2. Pull-Down Assay of MBP-Tagged Peptides

MBP pull-down was used to assess the binding of T1A, T1A-derived, and B1A peptides to ASK1, GST-MKK4/7, or His-JNK3α2. MBP-fusions of arrestin-3 and its peptides (10–30 μg in 50 μL of 20 mM Tris-HCl, pH 7.2, 150 mM NaCl) were immobilized on amylose resin (25 μL, 50% slurry, New England Biolabs, Ipswich, MA, USA) for 1 h at 4 °C with gentle rotation. Either GST-MKK4/7, His-JNK3α2, or ASK1 (5–10 μg in 50 μL of 20 mM Tris-HCl, pH 7.2, 150 mM NaCl) were added to MBP fusions immobilized on amylose resin and rotated gently for 2 h at 4 °C. The resin was resuspended and transferred to centrifuge filters (Durapore^®^-PVDF-0.65 μm, Sigma-Aldrich, St. Louis, MO, USA). Samples were quickly washed three times with ice-cold wash buffer (300 μL of 50 mM of HEPES-Na pH 7.3, 150 mM NaCl). Elution buffer (100 μL of 50 mM maltose in wash buffer) was added to each sample, and the tubes were rotated gently for 5 min at 4 °C before centrifugation. Protein in eluates was precipitated with 90% methanol, and pelleted by 10 min centrifugation at room temperature (Eppendorf 5418 tabletop centrifuge, 14,000 rpm). The supernatant was discarded, and the pellet was washed with methanol and allowed to air dry. The pellets were dissolved in sample buffer (30 μL Laemmli 2×), and eluates were analyzed by SDS-PAGE followed by Western blotting with primary rabbit anti-ASK1 antibody (Cell Signaling Technology #3762, 1:1000, Danvers, MA, USA), rabbit monoclonal anti-GST antibody (Cell Signaling Technology #2625), followed by anti-rabbit HRP-conjugated secondary antibody (Jackson ImmunoResearch, West Grove, PA, USA; 1:10,000).

### 4.3. Pull-Down Assay of His-Tagged JNK3

His pull-down was used to assess the binding of wild type and mutant arrestin-2/3 to His-JNK3. His-JNK3 (10 μg in 50 μL of 20 mM Tris-HCl, pH 7.2, 150 mM NaCl) was purified as described [20,27], and immobilized on the nickel-NTA resin (25 μL, Qiagen, Germantown, MD, USA) for 1 h at 4 °C with gentle rotation. Purified arrestin proteins (5 μg in 50 μL of 20 mM Tris-HCl, pH 7.2, 150 mM NaCl) were added to the immobilized His-JNK3 and rotated gently for 2 h at 4 °C. The resin was resuspended and transferred to centrifuge filters (Durapore^®^-PVDF-0.65 μm). Filters were quickly washed three times with ice-cold wash buffer (300 μL of 20 mM Tris-HCl pH 7.5, 150 mM NaCl). Bound proteins were eluted with 100 μL of 100 mM imidazole in wash buffer; the tubes were rotated gently for 5 min at 4 °C before centrifugation. Protein in eluates was precipitated with 90% methanol and pelleted by 10 min centrifugation at room temperature (Eppendorf 5418 tabletop centrifuge, 14,000 rpm). The supernatant was discarded, and the pellet was washed with methanol and allowed to air dry. The pellets were dissolved in sample buffer (30 μL Laemmli 2×), and eluates were analyzed by SDS-PAGE followed by Western blotting with anti-JNK (Cell Signaling Technology #9252; 1:1000) and anti-phospho-JNK3 (Cell Signaling Technology #9252; 1:1000) primary antibodies, followed by anti-rabbit HRP-conjugated secondary antibody (Jackson ImmunoResearch, West Grove, PA, USA; 1:10,000).

### 4.4. In-Cell JNK3 Activation Assay

Human embryonic kidney 293 (HEK293) cells, wild type or with CRISPR-mediated deletion of arrestin-2/3 (DKO) [30,31] (a generous gift of Dr. A. Inoue, Tohoku University, Sendai, Japan), were used. The cells were cultured in DMEM containing 10% heat-inactivated FBS (Invitrogen, Carlsbad, CA, USA) and penicillin/streptomycin (Gibco, ThermoFisher, Waltham, MA, USA) at 37 °C and 5% CO_2_. Cells were transfected at 80–85% confluency in 24-well plates with pcDNA3-HA-JNK3α2 and pcDNA3-Venus fusion constructs at a 1:3 ratio of Trans-Hi transfection reagent (FormuMax, Sunnyvale, CA, USA). At 48 h post-transfection, the cells were washed twice with cold PBS, and lysed using 50 mM Tris (pH 7.8), 2 mM EDTA, 250 mM NaCl, 10% glycerol, 0.5% NP-40, 2 mM benzamide, 1 mM PMSF, and protease/phosphatase inhibitors (Pierce). Protein concentration was measured using the Bio-Rad Quick Start™ Bovine Serum Albumin (BSA) Standard Set. The proteins were subjected to 10% SDS-PAGE and transferred to the PVDF membrane (Millipore, Bedford, MA, USA). Membranes were incubated with respective primary antibodies (Cell Signaling Technology, Inc., Danvers, MA, USA) anti-ASK1 (#3762S), anti-phospho-JNK (#9251), anti-SAPK/JNK (#4668P), anti-GST (#2625), or anti-HA (#3724), followed by the appropriate HRP-conjugated secondary antibodies (Jackson ImmunoResearch, West Grove, PA, USA). Bands were detected by X-ray film using enhanced chemiluminescence (ECL, Pierce, via Thermo Fisher Scientific, Waltham, MA, USA), and quantification was done using Quantity One software (Bio-Rad Laboratories, Hercules, CA, USA).

### 4.5. Data Analysis and Statistics

Statistical significance was determined with one-way ANOVA (analysis of variance) followed by Dunnett’s post hoc tests with correction for multiple comparisons using Prism8 software (GraphPad, San Diego, CA, USA). The values obtained with MBP (pull-down assays) or Venus controls (in-cell activation) were used as the comparison group for the Dunnett test.

## Figures and Tables

**Figure 1 ijms-23-08679-f001:**
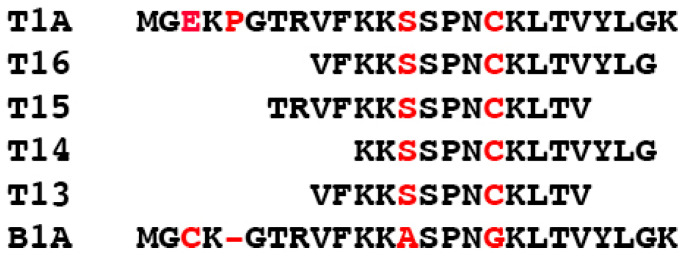
**Peptide sequences.** The sequence of T1A, shorter T1A-derived peptides, and B1A (homologous to the T1A element of arrestin-2). The residues that are different in arrestin-2 and arrestin-3 are shown in red.

**Figure 2 ijms-23-08679-f002:**
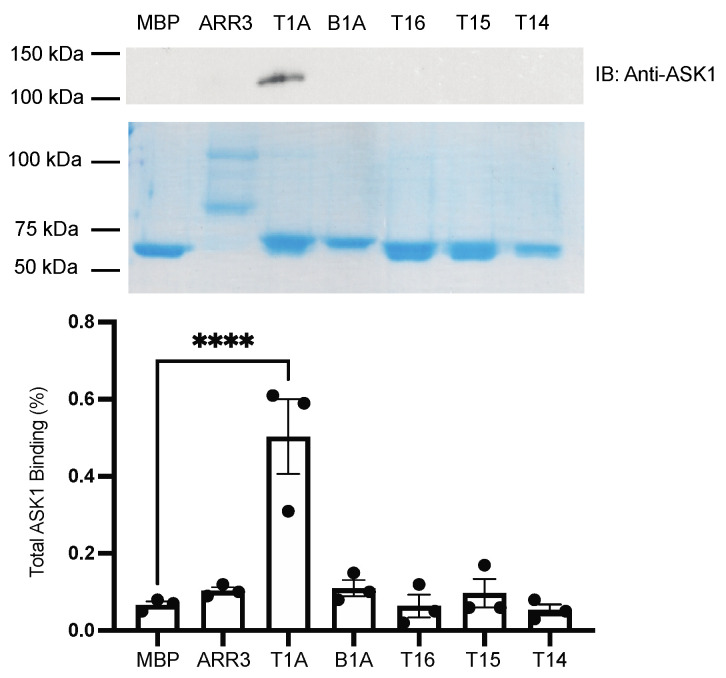
**MBP pull-down of ASK1.** Purified MBP fusions of indicated peptides and full-length arrestin-3 (10 μg) were immobilized on amylose resin and incubated with purified ASK1 (5 μg), as described in the methods. The beads were washed, and bound proteins were eluted with 50 mM maltose. Aliquots of eluates were subjected to SDS-PAGE. Coomassie staining (middle panel) was used to assess the loading of MBP constructs. Double bands of MBP-peptides and MBP-arrestin-3 here and in Figure 3, Figure 4 and Figure 5 likely result from proteolysis of purified proteins. ASK1 binding was measured by Western (top panel). Densitometric quantification (bottom panel) was performed using ImageJ software [26]. Dots represent measurements in individual experiments. Statistical analysis was performed using one-way ANOVA followed by Dunnett’s post hoc test with correction for multiple comparisons (*n* = 3). ****, *p* < 0.0001.

**Figure 3 ijms-23-08679-f003:**
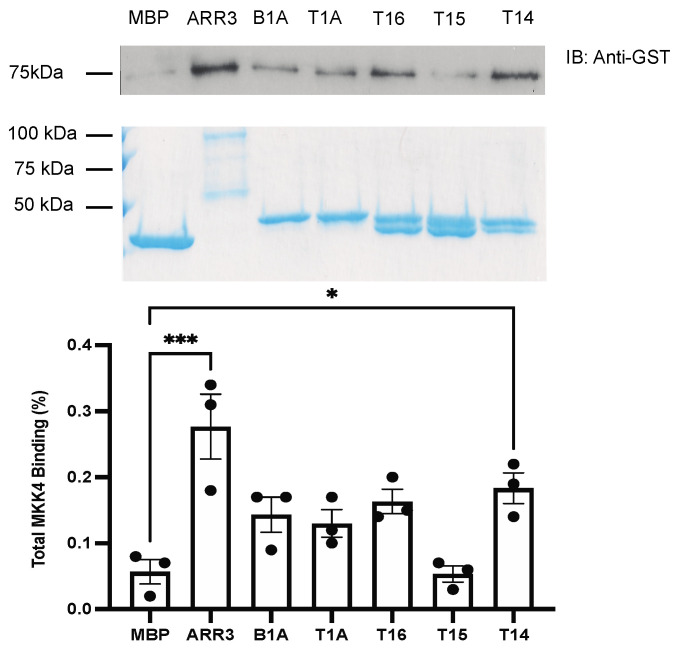
**MBP pull-down of GST-MKK4.** Purified MBP fusions of indicated peptides and full-length arrestin-3 (10 μg) were immobilized on amylose resin and incubated with purified GST-MKK4 (5 μg), as described in the methods. The beads were washed, and bound proteins were eluted with 50 mM maltose. Aliquots of eluates were subjected to SDS-PAGE. The loading of MBP constructs was assessed by Coomassie staining (middle panel). MKK4 binding was measured by Western (top panel). Densitometric quantification (bottom panel) was performed using ImageJ software [26]. Dots represent measurements in individual experiments. Statistical analysis was performed using one-way ANOVA followed by Dunnett’s post hoc test with correction for multiple comparisons (*n* = 3). *, *p* < 0.05; ***, *p* < 0.0005 to MBP control. Part of these data (the first four columns) was previously published in [21].

**Figure 4 ijms-23-08679-f004:**
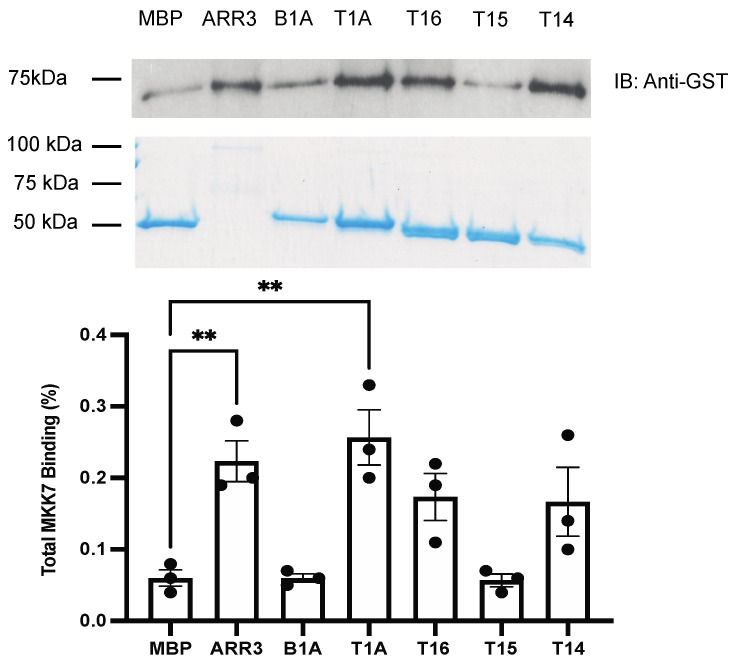
**MBP pull-down of GST-MKK7.** Purified MBP fusions of indicated peptides and full-length arrestin-3 (10 μg) were immobilized on amylose resin and incubated with purified GST-MKK7 (5 μg), as described in the methods. The beads were washed, and bound proteins were eluted with 50 mM maltose. Aliquots of eluates were subjected to SDS-PAGE. The loading of MBP constructs was assessed by Coomassie staining (**middle panel**). MKK7 binding was measured by Western (**top panel**). Densitometric quantification (**bottom panel**) was performed using ImageJ software [26]. Dots represent measurements in individual experiments. Statistical analysis was performed using one-way ANOVA followed by Dunnett’s post hoc test with correction for multiple comparisons (*n* = 3). **, *p* < 0.01 to MBP control. Part of these data (the first four columns) was previously published in [21].

**Figure 5 ijms-23-08679-f005:**
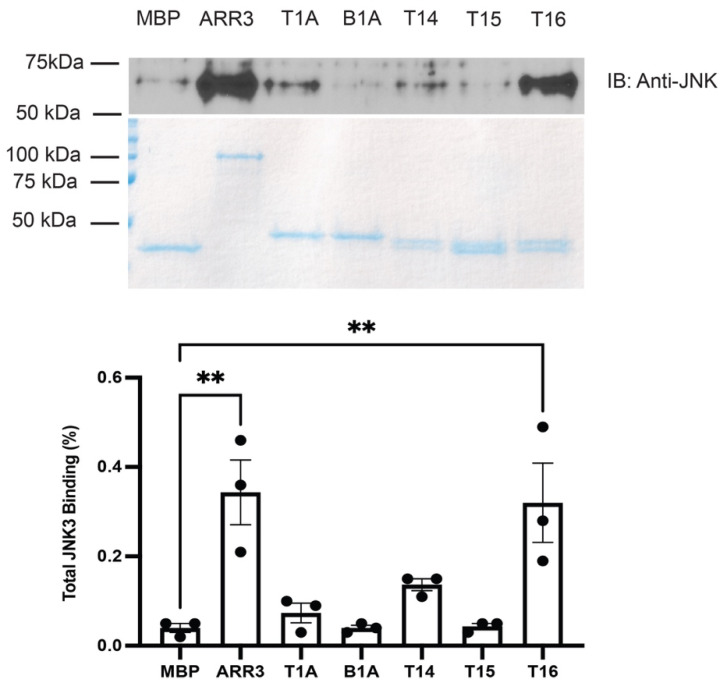
**MBP pull-down of His-JNK3α2.** Purified MBP fusions of indicated peptides and full-length arrestin-3 (10 μg) were immobilized on amylose resin and incubated with purified His-JNK3α2 (5 μg), as described in methods. The beads were washed, and bound proteins were eluted with 50 mM maltose. Aliquots of eluates were subjected to SDS-PAGE. The loading of MBP constructs was assessed by Coomassie staining (**middle panel**). JNK3α2 binding was measured by Western (**top panel**). Densitometric quantification (**bottom panel**) was performed using ImageJ software [26]. Dots represent measurements in individual experiments. Statistical analysis was performed using one-way ANOVA followed by Dunnett’s test with correction for multiple comparisons (*n* = 3). **, *p* < 0.01 to MBP control. Part of these data (the first four columns) was previously published in [21].

**Figure 6 ijms-23-08679-f006:**
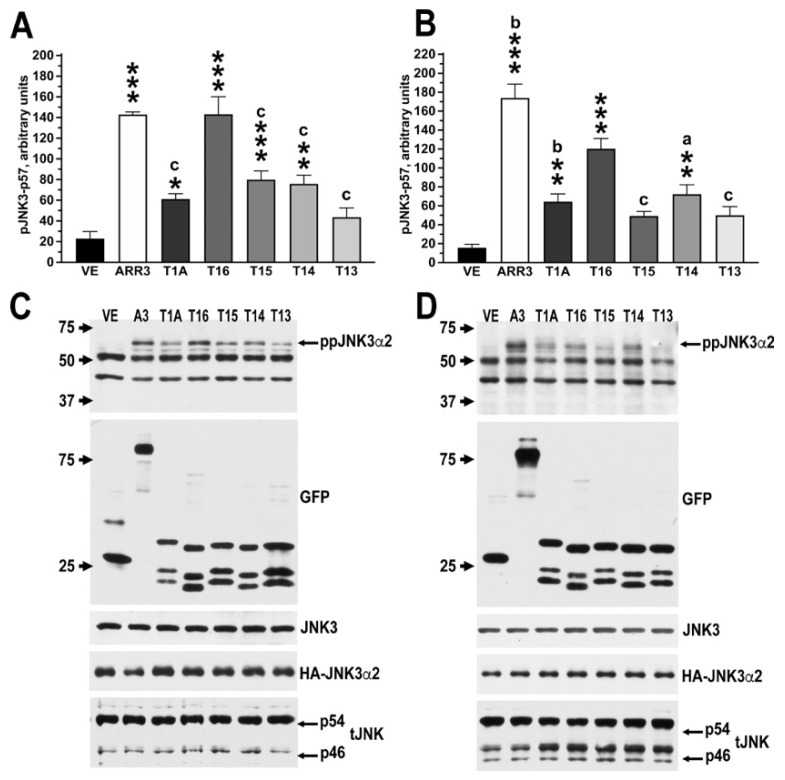
**In cell JNK3 activation.** (**A**,**B**) Quantification of the Western blot data of the peptide-induced JNK3 activation in HEK293 (**A**) or HEK293 arrestin-2/3 knockout (DKO) (**B**) cells. (**C**,**D**) Corresponding representative Western blots for phospho-JNK, total JNK, JNK3, HA, and GFP are shown for WT (**C**) and DKO (**D**) cells. The cells were transfected with HA- JNK3α2 (which is easy to detect because it runs higher than all isoforms of endogenous JNK1 and JNK2) and indicated Venus fusion of full-length arrestin-3 or peptide constructs. At 48 h post-transfection, the cells were lysed, the lysates processed and subjected to the Western blot analysis described in Methods. The band corresponding to the doubly phosphorylated ppJNK3α2 (indicated by arrows in (**C**,**D**)) was quantified by densitometry using Quantity One software. Westerns were used to demonstrate equal expression of HA-JNK3α2 using HA, total JNK, and JNK3 antibodies. Note that the total JNK antibody detects p54 and p46 bands, which contain eight JNK1/2 isoforms present in most cells [32], but does not detect the longer JNK3α2 isoform. All Venus fusions were detected with an anti-GFP antibody to compare the expression of different constructs. The Venus peptides run slightly above the Venus band; multiple lower molecular weight bands result from in-cell proteolysis. Statistical analysis was performed using Prizm9 software. The data were analyzed by one-way ANOVA followed by Dunnett’s post hoc test to compare all constructs with the Venus control. To compare the activity of T16 with other constructs, we used the Bonferroni post hoc test with correction for multiple comparisons. N = 5–7. *, *p* < 0.05; **, *p* < 0.01; ***, *p* < 0.001 to the Venus values; a—*p* < 0.05, b—*p* < 0.01, c—*p* < 0.001 to the T16 values.

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
