# Peer review of "Short Arrestin-3-Derived Peptides Activate JNK3 in Cells"

_ijms, 2022, doi:10.3390/ijms23158679_

Round 1

Reviewer 1 Report

Overall this manuscript takes a novel approach to examining the binding of arrestin protein peptides with targets arriving at the suggestion that key amino acid sequences function to regulate low/high binding affinity, thereby allowing for faster switching between targets. These results will be of  interest to those in the field studying JNK signaling, kinase activation and protein-protein interactions.

However, the results, as presented, do not seem to support the conclusions as strongly as the authors claim. Mainly, this is in the area of differences between the peptides binding of the different targets. The densitometry analysis does not seem to match the gel in all cases, and appropriate analysis of densitometric data between peptides was not done to support conclusions stating that differences between peptides are significant.

To make the conclusions as stated, appropriate analyses must support the data (see line by line comments below). In addition to these concerns, some other minor comments are below as well.

Line 22: need to add abbreviate maltose binding protein here (MBP) - the authors do not do it down below before using the abbrev exclusively (it's confusing to the reader when an abbreviation is not defined).

Line 37-38: “important for organ and tissue development and have been implicated in cancer” (lacks citations)

Line 82: “purify ASK1 and demonstrate that it binds with high affinity to T1A but only weakly to full-length arrestin-3 (Figure 2)” – This is very unclear how they arrived at this result since the gel doesn’t show any ARR3 in the ASK1 pulldown. Magnifying the gel image does not show any faint bands. How did the densitometry show that some was present? And in the densitometry the ARR3 does not appear to be significantly different from the MBP control. The authors need to do the analysis between those to show the difference is significant – since that is the claim.

Line 93: T16 results – same comment as with full length ARR3 – the gel data (as shown) does not support the claim.

Line 109: “Shorter peptides bind MKK7 kinases less avidly than T1A. T14 and T16 bind MKK4 comparably to T1A.” – The figures do not show statistical comparison between them on the blot – how do you know the difference between T1A and T14 is real? They only show comparison with MBP and ARR3. So how can it be claimed that the affinity is statistically significant between T1A and T14 or T14 and T16 when the analysis between them has not been performed?

Line 112-114: Similar to above comments - Need comparison analysis between 14-16 to show difference. Not convincing between 14 and 16 in either MKK - how do they get that conclusion? In MKK4 – difference between neg ctrl (B1A) and 14, and between 16 and 1A are supposed to be different?

Lines 115 – 117: “Full-length arrestin-3 demonstrates a robust binding, stronger than the best-binding short peptides (Figures 3,4).” This is not consistent with the MKK7 data in Figure 4 where ARR3 seems the same as the strongest peptides (and less than T1A). How do they justify the conclusion?

Figure 3: Coomassie blue stain – why do some of the peptides have a double band? Could that be confounding the results with the pulldown? Please explain.

Lines 144 – 145: “It demonstrated robust binding to both full-length arrestin-3 and T14 while exhibiting only a modest binding to T1A and T16 (Figure 5).”

This gel is not very convincing of the claim - T1A seems no different than neg ctrl or T15 - how is that considered modest binding? Densitometry needs to be analyzed – biologically relevant neg ctrl (B1A) seems to be more robust than T1A and T16. B1A binds but does not activate – this is confusing in the figure and not addressed in the text - needs additional explanation – especially with the JNK results.

Figure 6: The reference to total JNK (tJNK on the gel) is only made in passing but not defined on the gel – citing a paper explaining why total JNK includes p46 and p54 would be appropriate.

Author Response

Reviewer #1:

Comments: Overall this manuscript takes a novel approach to examining the binding of arrestin protein peptides with targets arriving at the suggestion that key amino acid sequences function to regulate low/high binding affinity, thereby allowing for faster switching between targets. These results will be of interest to those in the field studying JNK signaling, kinase activation and protein-protein interactions.

However, the results, as presented, do not seem to support the conclusions as strongly as the authors claim. Mainly, this is in the area of differences between the peptides binding of the different targets. The densitometry analysis does not seem to match the gel in all cases, and appropriate analysis of densitometric data between peptides was not done to support conclusions stating that differences between peptides are significant.

To make the conclusions as stated, appropriate analyses must support the data (see line by line comments below). In addition to these concerns, some other minor comments are below as well.

Major/Minor Criticisms:

  1. Line 22: need to add abbreviate maltose binding protein here (MBP) - the authors do not do it down below before using the abbrev exclusively (it's confusing to the reader when an abbreviation is not defined).

Thanks for pointing this out! The abbreviation was introduced when we first used it in the main manuscript (line 75).

  1. Line 37-38: “important for organ and tissue development and have been implicated in cancer” (lacks citations)

Sorry for overlooking this. Two citations were added (out of many possible).

  1. Line 82: “purify ASK1 and demonstrate that it binds with high affinity to T1A but only weakly to full-length arrestin-3 (Figure 2)” – This is very unclear how they arrived at this result since the gel doesn’t show any ARR3 in the ASK1 pulldown. Magnifying the gel image does not show any faint bands. How did the densitometry show that some was present? And in the densitometry the ARR3 does not appear to be significantly different from the MBP control. The authors need to do the analysis between those to show the difference is significant – since that is the claim.

Thanks for pointing this out! Indeed, pull-down revealed no detectable ASK1 binding to full-length arrestin-3. We changed the text to match the results of statistical analysis.

  1. Line 93: T16 results – same comment as with full length ARR3 – the gel data (as shown) does not support the claim.

Thanks for pointing this out! Indeed, pull-down revealed detectable ASK1 binding only to T1A, but not to full-length arrestin-3 or other peptides. We changed the text to match the results of statistical analysis.

  1. Line 109: “Shorter peptides bind MKK7 kinases less avidly than T1A. T14 and T16 bind MKK4 comparably to T1A.” – The figures do not show statistical comparison between them on the blot – how do you know the difference between T1A and T14 is real? They only show comparison with MBP and ARR3. So how can it be claimed that the affinity is statistically significant between T1A and T14 or T14 and T16 when the analysis between them has not been performed?

Thanks for pointing this out! We changed the text to match the results of statistical analysis.

  1. Line 112-114: Similar to above comments - Need comparison analysis between 14-16 to show difference. Not convincing between 14 and 16 in either MKK - how do they get that conclusion? In MKK4 – difference between neg ctrl (B1A) and 14, and between 16 and 1A are supposed to be different?

The reviewer is right. We changed the text to match the results of statistical analysis.

  1. Lines 115 – 117: “Full-length arrestin-3 demonstrates a robust binding, stronger than the best-binding short peptides (Figures 3,4).” This is not consistent with the MKK7 data in Figure 4 where ARR3 seems the same as the strongest peptides (and less than T1A). How do they justify the conclusion?

The reviewer is right. We changed the text to match the results of statistical analysis.

  1. Figure 3: Coomassie blue stain – why do some of the peptides have a double band? Could that be confounding the results with the pulldown? Please explain.

Thanks! Done in the legend to Figure 2 (lines 104-105). Double bands are likely the result of proteolysis. We noted this in revised legend.

  1. Lines 144 – 145: “It demonstrated robust binding to both full-length arrestin-3 and T14 while exhibiting only a modest binding to T1A and T16 (Figure 5).” This gel is not very convincing of the claim - T1A seems no different than neg ctrl or T15 - how is that considered modest binding? Densitometry needs to be analyzed – biologically relevant neg ctrl (B1A) seems to be more robust than T1A and T16. B1A binds but does not activate – this is confusing in the figure and not addressed in the text - needs additional explanation – especially with the JNK results.

Thanks! We also apologize for mislabeling of the peptides in Fig. 5. The labeling and the description of the results were corrected.

  1. Figure 6: The reference to total JNK (tJNK on the gel) is only made in passing but not defined on the gel – citing a paper explaining why total JNK includes p46 and p54 would be appropriate.

Thanks! Appropriate reference is included (line 194).

Reviewer 2 Report

The experiments carried out are performed with overexpression of proteins in HEK293 cell, while the physiological conditions of the cells/neurons are different. For this reason, I think it would be useful to demonstrate the activity of the most interesting peptides (TA1, T16 and T14 with the control peptides) in neuroblastoma cells. This because in these cells there is the physiological expression of JNK3 and if the peptide will activate endogenous JNK3 this will validate the ideas that the T16 peptide binds ASK, MKK4/7 leading to the real activation of JNK3 in cells. The best experiment will be in neurons, since JNK3 is the JNKs brain specific isoform, but I see the difficulty of overexpressing proteins in primary neurons, so I suggest Neuroblastoma cell line (i.e. SHSY5Y).

Otherwise, it is difficult to believe that 20 aa of TA1 can link subsequentially AK1, MKKs and JNK3 as proposed.

Author Response

Reviewer #2:

Comments: The experiments carried out are performed with overexpression of proteins in HEK293 cell, while the physiological conditions of the cells/neurons are different. For this reason, I think it would be useful to demonstrate the activity of the most interesting peptides (TA1, T16 and T14 with the control peptides) in neuroblastoma cells. This because in these cells there is the physiological expression of JNK3 and if the peptide will activate endogenous JNK3 this will validate the ideas that the T16 peptide binds ASK, MKK4/7 leading to the real activation of JNK3 in cells. The best experiment will be in neurons, since JNK3 is the JNKs brain specific isoform, but I see the difficulty of overexpressing proteins in primary neurons, so I suggest Neuroblastoma cell line (i.e. SHSY5Y).

The reviewer is correct in principle. However, we asked here only one question: do shorter than T1A peptides facilitate JNK3 activation? The manuscript contains an answer to this question. Transferring these experiments to neurons, or even another cell line (like SH SY5Y) that endogenously expresses JNK3 is not a trivial task. Unfortunately, SH SY5Y cells mostly express JNK3a1, not the longer JNK3a2 used here. JNK3a1 runs on the gel at the same level as four longer isoforms of JNK1 and JNK2. This precludes quantification of its phosphorylation without immunoprecipitation, which has its own caveats. Thus, for technical reasons we’d have to express longer JNK3a2 isoform in SH SY5Y, as we did in HEK293 cells. An important advantage of HEK293 cells is availability of this line with both non-visual arrestins knocked out, which enables the comparison of cells with and without endogenous arrestins. We would also like to point out that in all previous publications on this subject arrestin-3 dependent activation of exogenous overexpressed JNK3a2 was described. In fact, the only study where endogenous JNK family kinases we shown to be activated by arrestin-3 involved JNK1 and JNK2 (J Biol Chem. 2013 Dec 27;288(52):37332-42).

Otherwise, it is difficult to believe that 20 aa of TA1 can link subsequentially AK1, MKKs and JNK3 as proposed.

Thanks for drawing our attention to this issue. We were also surprised by this finding. However, in a fully extended conformation of the peptide each residue adds 3.8 Å. This makes the maximum length of 25 aa and 16 aa peptides 95 Å and 60 Å, respectively. For comparison, the longest axis in the structure of full-length arrestins is 70-75 Å. Thus, without folding constrains of the full-length protein, these peptides can be fairly large. We added this consideration to the discussion (lines 212-216). It is equally likely that the peptide only binds two components at any one time to facilitate signaling. At the moment we (or anyone else, for that matter) do not have tools to explore the exact molecular mechanism of scaffolding of JNK3 activation cascade by peptides or full-length arrestin-3. Therefore, we asked the question that can be addressed experimentally: can the parts of T1A peptide facilitate JNK3 activation in cells? We believe that we have answered that question here

Round 2

Reviewer 1 Report

The authors addressed comments from first review in a satisfactory manner to proceed with publication.